# Intratumoral Heterogeneity in Lung Cancer

**DOI:** 10.3390/cancers15102709

**Published:** 2023-05-11

**Authors:** Xiaodi Lv, Zixian Mao, Xianjun Sun, Baojun Liu

**Affiliations:** 1Department of Integrative Medicine, Huashan Hospital, Fudan University, Shanghai 200437, China; 22211220076@m.fudan.edu.cn; 2Pujiang Community Health Center of Minhang District of Shanghai, Shanghai 201114, China; yumao1203@sina.com; 3Institutes of Integrative Medicine, Fudan University, Shanghai 200437, China

**Keywords:** intratumoral heterogeneity, lung cancer, space and time, therapeutic strategy

## Abstract

**Simple Summary:**

The treatment of lung cancer poses a challenge due to the alterations in the characteristics of the same cancer cell. The altered characteristics make therapeutic strategies dynamic and evolutionary. To create a more appropriate therapeutic strategy for patients with lung cancer and improve their prognosis, we need to know how the alterations occur and how the alterations affect the treatment. This review aims to summarize the various variations in cancer characteristics and describe a more comprehensive multidisciplinary therapeutic strategy for lung cancer.

**Abstract:**

The diagnosis and treatment of lung cancer (LC) is always a challenge. The difficulty in the decision of therapeutic schedule and diagnosis is directly related to intratumoral heterogeneity (ITH) in the progression of LC. It has been proven that most tumors emerge and evolve under the pressure of their living microenvironment, which involves genetic, immunological, metabolic, and therapeutic components. While most research on ITH revealed multiple mechanisms and characteristic, a systemic exposition of ITH in LC is still hard to find. In this review, we describe how ITH in LC develops from the perspective of space and time. We discuss elaborate details and affection of every aspect of ITH in LC and the relationship between them. Based on ITH in LC, we describe a more accurate multidisciplinary therapeutic strategy on LC and provide the newest opinion on the potential approach of LC therapy.

## 1. Introduction

ITH is seen in most tumors during the investigation of the evolutionary trajectory of multiple cancer types. The factors affecting ITH in LC can be either genetic factors, pulmonary neoplastic microenvironmental factors, or metabolic factors. Both spatial and temporal ITH are influenced by the above three factors; therefore, both have emerged as typical characteristics in pulmonary neoplastic gene mutation, microenvironment, and metabolism.

While exploring the source of LC ITH, gene mutation was found to be the basis of spatiotemporal ITH in the pulmonary tumor microenvironment and LC metabolism. Recently, in a particular pathological type of lung cancer, remarkable genetic variation was found by multi-region genome-sequencing studies. The alterations were associated with not only anatomical position and stage of pulmonary neoplasms but also with distinct areas of the same tumor, which is often called spatial ITH. Additionally, the trait of the same nidus may vary prominently due to genetic factors and is referred to as temporal ITH. The dynamic headstream of variation of genetic factors is genetic alteration, such as gene mutation, which can occur in driver genes (can directly grant cancer cells superiority of survival and growth) and in passenger genes (cannot confer preponderance of selection). The form of gene mutation leading to ITH covers sequence insertion and deletion, single-nucleotide variants, and copy number variants [1].

From the perspective of therapy, the alteration of behavior correlated with the immunity of malignant pulmonary tumor should be seriously focused on. The characteristic immune response of pulmonary tumors can be recognized from the immunological ITH and tumor microenvironment (TME) ITH. Many elements in pulmonary TME vary both spatially and temporally, constructing a gigantic net of therapeutic schedules and pulmonary tumor ITH. In addition to the non-immune part, tumor immune microenvironment (TIME) ITH forms a landscape continuum based on spatiotemporal dimensionality, presented by affected immune cell function and immune factors. Multiple factors are secreted by pulmonary tumor cells and cells engaged in immune response progression.

Recognition of the relationship between metabolism and supervising gene changes in LC is a fundamental but inadequate process. Metabolic ITH is not only a factor affecting the variation of TME in LC but also a consequence of genetic ITH. Thus, the most appropriate dynamic detection of LC ITH can emerge from the perception of metabolic spatiotemporal ITH in LC. The acknowledgment of signal pathways associated with the alteration of metabolism and of the activation or suppression of typical key signal molecules is highly important. Two types of signal pathways exert a significant influence on metabolic ITH in LC, namely, phosphatidylinositol-3-kinase/protein kinase B/mammalian target of rapamycin (PI3K/AKT/mTOR) signal pathway and mitogen-activated protein kinases (MAPK) signal pathway.

In order to cure patients with LC and decrease their mortality, an integration concept should be formed by combining ITH with any therapeutic approach, no matter what mechanism the strategy is based on. In this review, this immature idea can be presented with new hypotheses and findings. More efforts should be made to transform this idea into different types of personalized and flexible treatment schedules in the hope of prolonging the survival time of every patient with LC.

## 2. Genetic ITH in LC

### 2.1. Gene Mutation as a Source of Genetic ITH

Driver mutation in non-small cell lung cancer (NSCLC) occurs in *EGFR*, *BRAF*, and *MET* and is exclusively clonal and early, whereas subclonal driver mutation can occur in most instances, such as in *PIK3CA*, *NF1*, *KRAS*, *TP53*, and *NOTCH* family members. Two types of driver mutations assume distinct duties in tumor generation and therapies. When certainly targetable, an exclusively clonal driver mutation can be the direct result of targetable treatment across multiple positions of disease, subclonal driver mutation always appears subclonally or is absent in most regions. The latter can be clonal in a single region, where a large fraction of subclonal driver mutations accumulate [2,3] (Figure 1). In LC, a high subclonal mutation burden has been reported to be closely associated with human leukocyte antigen (HLA) loss of heterozygosity (LOH).

In LC, the most frequent genes involved in pathogenesis in different pathological types have already been identified. For squamous cell carcinoma (SqCC) and adenocarcinoma (ADC), sequencing results have shown some differences; for example, the aberrations detected between ADC and SqCC are in *KRAS* (19% versus 2%), *TP53* (44% versus 69%), and *STK11* (21% versus 2%) [4]. For small cell lung cancer (SCLC), earlier findings had shown activating mutations of *EGFR* and *KRAS* and inactivating mutations of *TP53* and *Rb1* to be the most frequent [5]. Additionally, tumor-progressing factors related to genetic instability (mutations in DNA repair genes), treatment resistance, and antigenicity variation are of great importance [6].

Increasing evidence has shown *TP53* and *KRAS* are necessary for ITH and to present various pathogenic variants in different populations. For example, the mutation frequencies of *KRAS* and *TP53* in Iranian patients with LC are, respectively, lower and higher than in other populations [7]. In addition, *Q808H* in *DDR2*, *F212L*, and *D186G* in coding regions of *TP53*, three new putative pathogenic variants, have been first discovered in Iranian patients with LC [7]. The mutation frequencies of a consecutive population-based Swedish cohort have been proven to be similar to those observed in other Western populations, except for a notable high frequency (43%) of activating KRAS mutations among patients with lung adenocarcinoma [8]. In Western populations, a large fraction of ADCs harbors an activating mutation in KRAS [9]. Furthermore, the research showed patients who had the isolated KRAS mutation had a comparable overall survival rate to the wild-type group [10]. However, patients who had co-occurring mutations, specifically in TP53, had a poorer overall survival rate than both the wild-type group and the KRAS-only group [10]. Furthermore, a molecular subtype of *KRAS*-mutant LC with co-mutations in *KEAP1/NFE2L2* was identified to result in a significantly shorter overall survival. Patients with concurrent mutations in *KRAS* and *KEAP1/NFE2L2* had a shorter duration of therapy with platinum-based chemotherapy than other patients with *KRAS*-mutant lung cancer [10]. The mutation of either KRAS or TP53 had a tight connection with an enhanced PD-L1 expression [11,12]. A large and sustained clinical benefit was observed in KRAS^G12C^/TP53^mut^, associated with a higher share of the highest PD-L1 expression levels (≥90%: 41.7% vs. 20.0% in KRAS^other^) [13]. According to a large population-based case–control study conducted among Caucasians, Native Hawaiians residing on the island of Oahu, and Japanese, the substitution of arginine with proline at codon 72 of TP53 does not seem to perform a significant role in determining the risk of developing lung cancer [14]. The co-occurrence of TP53 and KRAS mutations was found to be associated with a higher mutation burden and was particularly enriched in the TH subset. This suggests that the dual mutation of TP53 and KRAS may have a synergistic and complementary effect on regulating immune biomarkers, leading to a responsive TME with adaptive immune resistance and increased immunogenicity [15].

Hence, the deepened exploration of LC gene mutation can help select appropriate therapy strategies. In preclinical studies with lung cancer cell lines expressing mutated TP53, the TP53 replacement strategy has improved chemotherapy and radiotherapy responses [16,17]. A meta-analysis of randomized controlled trials comparing combined therapy between immune checkpoint inhibitors (ICIs) and chemotherapy with chemotherapy alone showed that the addition of immune checkpoint inhibitors to chemotherapy may improve overall survival compared with chemotherapy alone [18]. Furthermore, the increasing evidence indicated that survival can benefit from adding immunotherapy (excluding checkpoint inhibitors) to conventional curative surgery or radiotherapy [19]. The application of ICIs, anti-PD1, and anti-PDL1 drugs has also been explored to select the best immunotherapy strategy for LC. A meta-analysis confirmed that the superiority of ICIs was over docetaxel in pretreated non-small-cell lung cancer patients and indicated a slight benefit from anti-PD-1 than from anti-PD-L1 inhibitors [20].

Gene mutation as the source of genetic ITH provided various signal targets for the development of therapeutic strategies and, meanwhile, made immunotherapy and other therapy strategies more flexible and specific to deal with treatment resistance due to mutation of various genes. Understanding the biology and natural history of LC is crucial for informing clinical management and therapeutic strategies. The discovery of significant ITH across various genomic types and similar tumor evolutionary trajectories for genetic changes has implications in this regard. Obtaining multiple biopsies and analyzing multiple genomic types may be necessary to capture the targetable events landscape accurately. Further studies are needed to identify the best combination of genomic types and regions to predict clinical outcomes, highlighting the importance of larger studies in the future.

### 2.2. Treatment Resistance Due to Specific Gene Mutation

When loss of phosphatase and tensin homolog (PTEN) occurs, resistance to PI3K inhibitors is seen. Activation of PIP3 phosphorylation accelerates the activity of protein kinase B, which is also called AKT. According to the activity of this central key enzyme, many pathways affecting cell survival, cell cycle progression, apoptosis, metabolism (such as protein synthesis, glycolysis, gluconeogenesis, and glucose uptake), cell proliferation, DNA repair, angiogenesis, and vesicle transport are likely to be suppressed or activated. Similarly, from the perspective of therapeutic resistance, owing to ITH in epidermal growth factor receptor (EGFR) or anaplastic lymphoma kinase (ALK), tyrosine kinase inhibitor (TKI) response rates are distinct across patients.

### 2.3. Antigenicity Variation Due to Specific Gene Mutation

Due to the loss of major histocompatibility complex (MHC) class I-coding genes in patients with lung cancer, pulmonary neoplasm can escape the scrutiny of the immune system [21]. Additionally, nonsynonymous mutations and indels in protein-coding genes create tumor neoantigens (TNAs) in addition to the normal antigens of pulmonary parenchyma cells and mesenchymal cells, which can be recognized and killed by tumor-specific CD8^+^ cytotoxic T lymphocytes (CTLs). According to previous clinical experience, TNAs due to gene mutation of *KRAS* in human gastrointestinal cancers have already been identified. It would be beneficial to harness the infiltration and cultivation of CTLs, in the course of treatment of tumors, which are triggered by *KRAS* mutation, which is considered one of the most malignant driver mutations. The prevalence of *KRAS* mutation can be high in most tumors, such as gastrointestinal cancers [22], as discussed previously for ADC.

### 2.4. Genomic Instability Due to Specific Gene Mutation

One type of gene mutation can influence both the possibility of genomic instability and the likelihood of TNA generation; it is called microsatellite instability (MSI), which benefits the extension or construction of short repetitive DNA sequences when DNA mismatch repair (MMR) genes appear to be defective [23]. MSI-high (MSI-H) pulmonary tumors are always associated with a high prevalence of generating TNAs, which can represent ITH in phenotype and be recognized by the specific immune system [24].

### 2.5. Relationship between Genetic ITH and Immunological ITH

Immunological ITH can be of three forms, namely, antigenicity, adjuvanticity, and immunoevasion [25]. Meanwhile, the neoplastic microenvironment can be changed by means of metabolic flexibility.

Immunological ITH in lung cancer is the manifestation of genetic ITH, according to the crucial degree of transcriptional ITH, of genes linked to immunity in spatial and temporal dimensions.

As a result, a huge stress exists on selection from the specific immune system, impacting the lung cancer cell clonal architecture, which has emerged as cancer clones or subclones. Lung cancer cells take advantage of the preceding mechanisms to escape the supervision and effect of the immune system. For instance, in lung cancer evolution, antigen expression can be suppressed in the form of limited antigenicity downstream, or antigen presentation can be impaired due to defects in molecular mechanisms [26,27]. Alterations can also be found in lung cancer cell adjuvanticity and in the establishment of a highly immunosuppressive TME [28,29].

## 3. ITH of Pulmonary Neoplastic Microenvironment

### 3.1. Immunological ITH and Pulmonary Neoplastic Microenvironment

Immunological ITH is one of the most principal factors affecting the neoplastic microenvironment, whereas the change in neoplastic microenvironment can accelerate or suppress the progress of genetic ITH, thereby influencing immunological ITH.

### 3.2. Other Neoplastic Microenvironmental Factors and TMH

Partial alterations or temporal variations in other TME factors, such as vascularization, stiffness, and inflammation, make a considerable contribution to neoplastic microenvironmental ITH [30].

ITH in TME (fused as tumor microenvironment heterogeneity (TMH)) provides a pulmonary tumor with the capability of drug resistance. Multiple growth factors and cytokines in TME induced by therapy can reprogram stromal cells toward protumor or antitumor phenotypes [31].

### 3.3. TIME ITH Affected by TME and Genetic ITH

The acknowledgment of the diversity of TIME (part of TME) due to TME factors and immunological ITH is of great significance so that new therapeutic targets can be found and their roles in patients receiving immune check-point blockade (ICB) therapy can be validated. The therapy is deemed to be a hindrance to the co-activities of the receptor and/or the ligand, such as cytotoxic T lymphocyte-associated antigen-4 (CTLA-4) and programmed cell death protein (PD-1). Recently, it has been found that vitamin E(VitE) can increase the antitumor efficacy of immune check-point therapy (ICT) (Figure 2). In this way, ITH occurs in TIME when protein tyrosine phosphatase SHP1 in dendritic cells (DCs) are inhibited. Then, DCs can enhance different levels of cross-presentation of tumor antigens, and different numbers of extracellular vesicles of DCs can trigger systemic antigen-specific T cell antitumor immunity in various degrees (Figure 2) [32].

In lung cancer, the source of TIME variation must be genetic ITH (Figure 3). The main pattern of manifestation of TIME ITH is in the alteration of cytokine production. As reported previously, cytokines whose expression is regulated by oncogenes are indispensable for the recruitment and phenotype of immune cells, especially myeloid immune cells. In advanced lung adenocarcinoma cases, *KRAS G12D*-triggered pancreatic ductal adenocarcinoma (PDAC) has been repeatedly demonstrated to secrete a large amount of granulocyte monocyte colony-stimulating factor (GM-CSF), suppressing immune function partially by improving the number of tumor-associated Gr-1^+^ CD11b^+^ myeloid cells. The immunosuppressive TIME established by the oncogene supports the development and proliferation of a variety of malignant tumors in addition to ADC, such as pancreatic neoplasia [33,34].

Furthermore, the secretion of tumor-derived chemokines, induced by specific oncogenes, is also important for the generation of TIME ITH (Figure 3). As is revealed by transcriptional analysis of tumor cells and in vitro DC migration assays in a mouse model of NSCLC with *BRAF* V600E mutation and *PTEN*-deficiency, reduction in chemokine CC motif ligand4 (Ccl4) production, which is deemed to be an efficient chemoattractant for multiple kinds of myeloid cells, including CD103^+^ DCs, could explain the undesirable infiltration of CD8^+^ T cells into TIME induced by the declined recruitment of CD103^+^ DCs.

Humoral factors in TME have also been confirmed to be momentous elements regulating TIME (Figure 3). For instance, pentraxin3 (PTX3) from mice, a significant regulator activating complement by means of reciprocity with factor H, can mediate monocyte recruitment and tumor-associated macrophage (TAM) phenotype to fulfill the purpose of repressing tumor growth [35]. Therefore, human PTX3 might affect the establishment of TIME similarly by the revelation of PTX3 promoter hypermethylation.

Tumor extrinsic factors can also perform a crucial role in triggering the profound change in the structure of TIME. For example, the non-immune stromal ingredient of TME can influence immune components. In NSCLC, *BRAF*V600E promotes the generation of interleukin (IL), such as IL-1α and IL-1β, hence repressing the recognition and killing capacity of tumor-specific CTLs, partially through the upregulation of PD-1 ligands, PD-L1, and PD-L2, and via secretion of cyclooxygenase (COX-2). A similar mechanism was also found in melanoma [36].

### 3.4. Immunoescape Affected by TIME

Immunoescape can also be facilitated by genetic ITH promoting the alterations in TIME. Genetic variations can occur due to mutations in genes regulating genomic immovability, such as MYC proto-oncogene (*MYC*), basic helix-loop-helix (bHLH) transcription factor, KRAS proto-oncogene GTPase (KRAS), and tumor protein p53 (TP53). Modification of these genes can elevate the abundance of immunosuppressive cell infiltration and/or organize the expression of coinhibitory molecules, such as PD-L1 or CTL exclusion. Genetic modifications can also make a difference in the differentiation of immune cells, such as of catenin beta 1 (CTNNB1), activation of which can exclude T cells [37].

### 3.5. Relationship of Immunoediting and ITH in LC

Cancer immunoediting is a dynamic process including three phases: elimination, equilibrium, and escape, whereby the immune system can not only protect against the development of tumors but also shape the cancer characters. A central principle of cancer immunoediting is associated with whether the T cell recognition of tumor antigens can promote the immunological elimination or sculpting of developing cancer. It has been proved that checkpoint blockade, such as the blocker targeting the PD-1/PD-L1 pathway, can effectively enhance T-cell-dependent immune-selective pressure. In addition to adaptive immunity and particularly T cells, innate immunity also performs a crucial role in tumor immunogenicity editing [38]. Hence, it can be concluded that immunoediting can produce variants with reduced immunogenicity, through which immunoediting facilitates immunological ITH.

Elimination is a phase of cancer immunoediting where the immune system can detect and destroy varieties of early tumors. The balance of tumor immunity is towards antitumor due to the upregulated expression of tumor antigens, FAS, TRAIL receptors, MHC class I on tumor cells and granzymes, IL-1, IL-12, TNF-α, IFN-α/β/γ, and perforin in TME. Therefore, the function of immunoediting also has a tight connection with TME. If TMH accelerates (such as an elevated spatial ITH of tumor-infiltrating lymphocytes (TILs) in LC), accumulated subclonal neoantigens mutations under the more immunoexhaustive and suppressive TME will contribute to more probability for tumor evasion [39]. The immunological and TME determine the destiny of both immunoediting and immune ITH. Tumors with low immune ITH can also escape, although the condition must be rigorous, including loss of HLA-LOH and immunoediting due to the high immunoselective pressure [40].

In the equilibrium phase of cancer immunoediting, the immune system sustains the tumor in a state of functional dormancy. The mechanism of this phase dynamically describes the development of tumor cells’ genetic and immunological ITH due to constant TME pressure. Tumor cell variants evolve towards the resistance of immune recognition (antigen defects and loss in antigen presentation) and enhancement of immunosuppression. In this phase, TME presents a balance between tumor-promoting cytokines (IL-10, IL-23) and antitumor cytokines (IL-12, IFN-γ). When NK cells and cytokines, such as IL-17A, IFN-α/β, and IL-4, are dispensable, the adaptive immune system will be required to involve in the maintenance of the functionally dormant state.

The mechanism of immune escape is fundamental for targeted immunotherapies. In this phase, tumor cells evade recognition of the immune system by loss of tumor antigens, co-stimulatory molecules or MHC class I, secrete cytokines TGF-β, IL-6, M-CSF, and VGEF, that increase angiogenesis and express molecules of immunosuppression (TDO, IDO, galectin-1/3/9, CD73, CD39, and PD-L1 adenosine receptors), survival (anti-apoptotic molecule bcl-2), and enhanced resistance (STAT-3). Furthermore, M2 macrophages, DCs, and myeloid-derived suppressor cells (MDSCs) can suppress the proliferation of CD8^+^ T cells or induce apoptosis by secreting IL-10 and TGF-β and express immunoregulatory molecules such as iNOS, IDO, and arginase. Regulatory T cells (Treg) can be generated by IDO-expressing DCs or MDSCs. T cells, including Treg cells, can express inhibitory receptors, such as LAG-3, Tim-3, PD-1, and CTLA-4, suppressing the antitumor immune response and promoting tumor outgrowth. In this phase, the balance is skewed toward tumor progression resulting from the immunosuppressive molecules and cytokines, such as PD-L1, IDO, VEGF, TGF-β, and IL-10. To dynamically recognize the ITH of TIME and appropriately combine blockers targeting different pathways closely associated with immunoevasion can revolutionize thinking about the therapeutic strategy of LC and approach to the essence of the pulmonary tumor.

## 4. Metabolic ITH in LC

### 4.1. Relationship across Metabolic ITH, Genetic ITH, and TMH

Metabolic remodeling has been regarded as a momentous manifestation in the process of genesis and development of tumors; the same is true in lung cancer. With mutations occurring in specific genes, the metabolic pathways can vary at the same pace, altering the components of TME to some extent. On the one hand, modification of the elements in TME due to the molecular mechanisms of metabolic remodeling can be an indispensable factor affecting TIME, according to the varieties of metabolites. On the other hand, alteration of TME can change the pulmonary neoplastic adaptability of proliferation and differentiation environment. Consequently, as per the theory of natural selection, the most adaptive tumor cells in different localities, according to metabolic division, with a specific gene mutation, would continuously proliferate and differentiate, promoting the generation and advancement of heterogeneity in the same type of LC.

The latest survey has revealed that variation in metabolic state can be an indispensable source of genetic ITH. Malic enzymes (ME), as sources of NADPH, accelerate the combination of NADPH and histone deacetylases 3 (HDAC3) in cancer cells and adipocytes, and influence the level of epigenetics. Additionally, ME2 can enhance the stability of mutant p53 (mutp53) by increasing the generation of 2-hydroxyglutaric acid (2-HG), through which the degree of ubiquitination and degradation of mutp53 can be decreased, mediated by murine double mimute2 (mdm2) [41]. Moreover, ITH in epigenetics also has a tight connection with metabolic ITH, in addition to the crucial role performed by HDAC3. Another enzyme called methyltransferase-like 3 (METTL3) can regulate the level of methylation of mRNA as a kind of methylase. The different expression of METTL3 increases the levels of expression of critical genes regulating the autophagy pathway, such as autophagy-related gene -5 (ATG-5) and ATG-7, and has proven that alterations in autophagy, mediated by METTL3, can present different degrees of gefitinib resistance in NSCLC, the mechanism of which is possibly connected tightly with temporal ITH (Figure 4) [42]. It has been proved that autophagy induction following hypoxia has been proposed as a survival mechanism driving tumor progression [43].

### 4.2. Main Forms of Metabolic ITH in LC

The processes of glucose and lipid metabolism in the development of LC are unlike that in normal cells. Metabolic reprogramming, identification of molecules in the reaction processes, such as glycolysis and liquid metabolism (Figure 5), and exploration of fundamental molecular mechanisms are, therefore, of great significance for clinical therapy.

### 4.3. ITH in Glucose Metabolism

In LC tissue, more glucose is harnessed for participation in the tricarboxylic acid cycle (TCA), and glycolysis rates and mitochondrial abilities vary in different LC cells [44]. On the one hand, highly glycolytic lung cancer cells provide the incremental amount of lactate, which can accelerate the immunosuppressive effects [45]. Along with the alteration of time and space, different metabolic landscapes provide distinct living pressure to tumors and hence promote ITH in LC. In NSCLC, superior manipulation of diverse carbon resources contributes to the heterogeneous configurations of glucose metabolism [46]. Moreover, fast and slow multiplying cell subpopulations have been found to coexist, respectively exhibiting a glycolytic metabolism corresponding to high chemosensitivity or an oxidative metabolism corresponding to chemoresistance. A similar molecular mechanism has been found in glioblastoma [47]. On the other hand, lung cancer cells can take advantage of different quantities of glucose for proliferation and development, according to the anabolic metabolism [48]. TCA, a key mediator, provides many intermediate products, satisfying the needs of proliferation and incursion of LC cells. As concluded by Warburg, defects in mitochondrial function can dysregulate the expression of key enzymes in the respiratory chain, destroy the electron transmission chain, and allow irregular expression of mitochondrial genes [49,50]. In LC, distinct levels of ROS, released from damaged respiratory chains, can repress the activity of key enzymes in TCA, such as aconitase, so that critic acid in mitochondria can accumulate step by step and then be disintegrated into acetyl-CoA and acetoacetic acid. The product is considered a fundamental material for cholesterol and fat synthesis, and the cycle is named the truncated TCA cycle [51]. Hence, ITH in glucose metabolism can contribute to the formation of ITH in lipid metabolism. Furthermore, overexpression of the key enzymes, namely, hexokinase 2 (HK2), phosphofructokinase (PFK), pyruvate kinase (PKM), and lactate dehydrogenase (LDH), in LC can facilitate LC cell proliferation via the AKT signaling pathway [52,53]. Metabolic reconstruction of LC can be an essential enhancing factor of drug resistance of epidermal growth factor receptor, tyrosine kinase inhibitor (EGFR-TKI) [54].

ITH in glucose metabolism can be regulated by exosomes with miR-122 from cancer cells. Exosomes from breast cancer cells (BCC) can suppress glucose metabolism by downregulating the glycolytic enzyme pyruvate kinase of premetastatic niche cells in lungs and hence promote metastasis. This mechanism provides a way to reduce metastasis by the systemic intervention of miR-122. It has been proven to be an effective approach to alleviating cancer-induced glucose recollection [55].

The capability of immune evasion and resistance to PD-1 are associated with ITH in glucose metabolism. The key enzyme pyruvate dehydrogenase complex E1α (PDHE1α) can be activated by oncogenic signaling. After phosphorylation, subcellular translocation of PDHE1α promotes tumor immune evasion and resistance to PD-1 [56].

### 4.4. ITH in Lipid Metabolism

The more activity due to endogenous fatty acid metabolism, the less the EGFR expression. Therefore, recognition of the fatty acid pathways that can be a new probable target of lung adenocarcinoma treatment would be worthwhile [57]. Additionally, the metabolism of endogenous fatty acids has been reported to be related to epithelial-mesenchymal-transition (EMT) management. With alteration in LC cells due to EMT, lesions in different regions would acquire various degrees of invasion abilities and metastatic capacities [58]. ITH of endogenous fatty acid metabolism can provide a source of lipid signals with partially diverse quantity and reactiveness, such as lysophosphatidic acid (LPA), and prostaglandin E2 (PGE2). Lipid signals, such as LPA, sphingosine-1-phosphate (S1P), and PGE2, have varieties of abilities, including altering the TIME by means of attracting immune cells and macrophages, and regenerating the capillaries of cancer cells [59,60]. PGE2 is a significant signal substance stimulating alternatively activated macrophage (which can be abbreviated as AAM and be called M2) polarization induced by Krupple-like factor 4 (KLF4) and cyclic AMP-responsive element binding (CREB). Therefore, PGE2 could weaken the supervision and activation of macrophages acting on pulmonary tumors, hence resulting in the escape of tumor cells from immune monitoring. Phospholipids, as one of the most crucial terminal products of endogenous fatty acid metabolism, have a momentous influence on protein modifications after translation, such as acetylation of proteins. In this manner, multiple indispensable signal pathways affecting cellular survival, growth, proliferation, development, cycle progression, and angiogenesis can be altered both spatially and temporally. The lipid raft structure formed by phosphatidylserine, phosphatidylinositol, and lecithin performs an essential role in a variety of vital signal pathways, such as rat sarcoma (RAS), wingless/integrated (Wnt), and P13K/AKT [61].

### 4.5. Typical Signal Pathways Influencing Metabolic ITH

Metabolic ITH has a tight connection with genetic ITH and is affected by many typical signal pathways (Figure 6). PI3K/AKT/mTOR signal pathway and mitogen-activated protein kinase/extracellular regulated protein kinases/adenosine 5ʹ-monophosphate-activated protein kinase (MEK/ERK/AMPK) signal pathway both participate in the metabolic remodeling of pulmonary tumor cells. When each typical signal pathway presents different levels of expression, the characteristics of the tumor get altered. The EGFR signal pathway has been shown to perform a critical role when tumor cells acquire anoikis resistance in peripheral blood, followed by metastasis [62]. Therefore, some therapeutic methods can be based on inhibiting signal pathways, which present oncogenic function. For example, inhibiting the expression of the mTOR signaling pathway and PCNA (proliferating cell nuclear antigen) using microRNAs is a good way to hinder the proliferation of pulmonary tumors. When PCNA expression is downregulated, the proliferation of circulating tumor cells (CTCs) can be inhibited [63].

Many drugs with obvious resistance to LC modulate ITH in metabolism and TME by the typical signal pathways, such as glycyrrhizin and glycyrrhetinic acid [6].

## 5. Spatial and Temporal ITH in LC

### 5.1. Temporal ITH in LC

Temporal ITH in LC has been studied for many years since it is remarkably essential for the comprehension of the reappearance and drug resistance of pulmonary tumor cells during a therapeutic schedule (Figure 7). The most apparent temporal ITH between primary pulmonary tumors and their metastases can be distinguished by concordance rate; meanwhile, there are many other applications of concordance or discordance, such as reviewing the effect of the methods applied in diagnosis and preservation, or estimating whether sample issues can be the proportion of tumor or not [64]. The concordance rate of EGFR between primary tumor and metastasis can range from 100% to 72%. The least proportion was measured by Schmid’s research group [65], while the highest one was reported by Shimizu’s investigation team [66]. Similarly, the concordance rate of another significant signal molecule KRAS has been found to fluctuate between 100% and 64%. The least concordance rate of KRAS between primary tumor and metastasis was identified by the Badalian G research group [67], the highest rate being measured by the Rosell R research group and the Alsdorf WH team [68,69].

#### 5.1.1. Lung Cancer Reappearance Due to Genetic Temporal ITH

Lung cancer temporal ITH caused a series of poor prognoses and high mortality problems. The reappearance after periods of treatment schedule is one of the most essential mechanisms of LC-related death. Reappearance mechanisms are mainly connected with the subclones of specific driver mutations, which can tend to have the pulmonary tumor evolving increasingly tolerant of the surrounding TME alteration due to the successive therapy. This event can also be regarded as a combination between genetic ITH and selective pressure. In addition, affecting factors on reoccurrence mechanisms include attainment of EMT phenotype, DNA hypermethylation, EGFR reverse bypass, synergetic interfaces with the target gene, and signal transduction-redundant activation [70,71,72].

#### 5.1.2. Drug Resistance Due to Genetic and TME Temporal ITH

After chemotherapy and careful census, some authors reported that the reaction rate of EGFR TKIs is much lower than that in pre-treatment conditions, and the mutation rate of gene *EGFR* obviously declined [73,74,75]. Overall survival, median survival time, and evolution of time after treatment with EGFR TKIs have been reported to be much lower in patients with LC, presenting absolutely obvious temporal ITH [76].

In LC, the regulation of biological processes associated with TMH involves a complex interplay between regulatory molecules and the microenvironment. This intricate system facilitates dynamic cellular communication through various mechanisms, such as indirect extracellular signaling, direct contact, and the exchange of extracellular vesicles (EVs). The secretome, which comprises cytokines, growth factors, hormones, enzymes, and EVs within the extracellular matrix, performs a crucial role in tumorigenesis by mediating multifaceted mechanisms that contribute to drug resistance [77,78,79,80].

#### 5.1.3. A Landscape Continuum of Functional Cells Due to Temporal ITH in TME

As mentioned above, the pulmonary neoplastic microenvironment (or TME) can present obvious heterogeneity in the form of alteration of immune compartments, which can also be called TIME, and variation of stromal compartments. Thus, the evolution of TME in LC upon treatment or natural conditions is closely related to the outcomes and prognosis of the disease [29,81]. The presentation of TME is a high-level ITH influencing TILs, DCs, TAMs, and cancer-associated fibroblasts (CAFs). Apparent alterations of TILs, DCs, and TAMs are representatives of the high degree of TIME ITH, whereas the variations of CAFs can be an index for measuring the heterogeneity of stromal compartments. The deeper sight into the cellular architecture of pulmonary tumors to find TMH is supposed to be provided by numerous single-cell RNA-seq and mass-cytometry studies [82]. T cells can be divided into subgroups from different perspectives according to varieties of standards. For example, T cells subgroups can be divided by the differentiation stage into naive T cells versus effector T cell (TEFF) and Treg cells, by the proliferative ability to cycle T cells and resting T cells, by the expression of effecter signal molecules into mainly secreting interferon gamma (IFN-γ) and others, by the clonality (which is the comparative quantity of a particular T cell clone among a T cell subset), by the metabolic profile into oxidative and glycolytic. In general, the characters mentioned above are indispensable standards to define a specific T cell with exact function from kinds of dimensions, and finally, become parameters capable of supporting or suppressing tumor growth. Only two opposite function states cannot completely conclude all features of the dynamic behavior of T cells. Hence, it is remarkably necessary to construct a continuum with stages of gradual change by highly plastic functional phenotypes, which present spatiotemporal ITH. For example, a continuum of functional stages about effector activity is used to measure the exhaustion of tumor-infiltrating CD8^+^ T cells [83,84]. Likewise, even silent and dysfunctional T cells can perform a high level of temporal heterogeneity in LC, where they can present different rates of proliferation and clonal expansion whether the treatment schedule is performed or not. Although immunotherapy research has focused largely on the ITH of T cells, the ITH of tumor-infiltrating B cells and plasma cells (TIL-Bs) also has been proven to perform a synergistic and crucial role in tumor control. TIL-Bs can potentially combat immune editing and tumor ITH through the easing of self-tolerance mechanisms. In addition, temporally functional alteration of TIL-Bs can change the capability of antigen presentation. TMH can be accelerated with the decreasing number of myeloid cells, NK cells, and T cells. In LC, TIL-Bs can present strong prognostic and predictive significance in the context of ICB and standard treatments, providing the prospect of new treatment strategies through their unique immunological properties [85].

There is another heterogeneous population of immune cells functioning as the induction of immune tolerance in TME called TAMs. TAMs can maintain the immunosuppressive TME by inducing M2 phenotypes generation in TAMs via PD-L1/PD1 signaling from LC or by attracting MDSCs, Tregs, and tumor-associated neutrophils (TANs). TAMs are functionally and phenotypically diverse beyond the M1/M2 dichotomy.

The alternative activation of TAMs has a tight connection with TME. On the one hand, TAMs in well-oxygenated areas show some qualities of classical (M1) activation, and the hypoxic microenvironment triggers an alternative activation of TAMs. On the other hand, hypoxic tumor cells produce cytokines, such as oncostatin, TGF-b, or IL-6, which provoke TAMs’ alternative (M2) activation. This activation promotes tumor progression and inhibits antitumor immune responses by blocking T-cells and NK cells. The release of HMGB-1 from tumor cells stimulated by hypoxia fosters IL-10 production in macrophages, which also supports an alternative activation. Additionally, tumor cells’ altered metabolism leads to the production of high levels of lactate. The activation of hypoxia-inducible factors (HIF-1 and HIF-2) in macrophages regulates genes that encourage tumor cell proliferation, angiogenesis, and metastasis, thereby favoring tumor progression. TAMs can also produce matrix metalloproteinases (MMPs), which perform important roles in the tissue remodeling associated with protein cleavage, modifying the immune microenvironment, local invasion, metastasis, angiogenesis, and tissue repair. Therefore, it is crucial to recognize the ITH of TAMs so that we can conclude the prognosis of patients with LC more accurately, regulate TAMs towards the antitumor activity and control the TMH, including both TIME and stromal compartment [86]. For clear and accurate recognition of TME, it is fundamental to know large-scale continuum covering different proportions of varieties of immune cells divided into types of subsets. Temporal heterogeneity in scales of different immune cell states around pulmonary tumor cells is supposed to be a crucial factor in the management of immune responses to LC resistance.

Generally, in the pulmonary neoplastic microenvironment, principal alterations of cell constituent included reduction in CTLs and DCs, collection of TREG cells, expansion of CAFs, intensified exhaustion of T cells, accumulation of immunosuppressive cells compartments, such as MDSCs and specific CAF subsets [87].

It has been proven that modification of tumor-stromal interactions performs an indispensable role in promoting temporal TMH where extracellular matrix (ECM) remodeled (termed as desmoplastic stroma) with the cleavage of adhesion proteins, such as integrins and cadherin, which can regulate migration and attachment of cancer cells and with the processing of intercellular junction proteins (desmoglein (Dsg)/desmocollin (Dsc)) which can control tumor cohesion and with the alteration of intercellular signaling. ECM proteins can be modulated by the regulation of plasmin, MMPs, uPA, and meprins operated by kallikrein-related peptidases (KLKs) and promote the modification of ECM composition, production of angiostatin, activation of pro-factors (HGF, EGF, and TGF-β) and release of growth factors. TMH occurs with the temporal variation of CAFs, immune cells, and endothelial cells in the primary tumor site due to the ECM remodeling with the changes to ECM cross-linking and tissue stiffness mediated by tumor-secreting KLKs [88,89]. ECM can also affect ITH from the perspective of regulating the tumor migration ability. When cancer cells migrate through ECM, the latter acts as a mechanical barrier to discern differences in the physical properties of cells, such as their size, shape, and rigidity. The collective tumor cell invasion involves ensembles of cells moving as ‘leaders’ and ‘followers’ and has been proposed to require greater cell–cell adhesion and matrix degradation as requisites for migration. However, even mesenchymal cells, with decreased expression of the epithelial markers, such as E-cadherin, are known to migrate collectively [90]. Cancer cells show different behaviors in distinct ECM milieus. For example, the response of cancer cells to chemotherapy has been reported to be regulated by substrata of different ECM molecules or their combinations (including, but not limited to collagen I, IV, and VI; laminin; and fibronectin). In addition, apoptosis of cancer cells can be modulated by ECM in a FAK- and β1 integrin-pMLC-YAP signaling-dependent manner [91]. Tumor migration and chemoresistance are sensitive to the nanospacing of the ECM ligands that engage their cognate receptors on the cell surface, suggesting that such plasticity is linked to dynamical changes in signaling landscapes [92].

TME can be reprogrammed by the modulation of CAF and tumor cells crosstalk, which can be altered towards the environment supporting tumor growth. The modulation of the protease-activated receptors (PARs) localized on both peri-tumoral stromal cells and cancer cells presents spatiotemporal ITH due to the varieties of proteolytic cleavage by KLKs and becomes potential targets to regulate TMH and control pulmonary tumor [93,94]. The immune contexture of TME can also alter with the change of ability of T cell response, activation of ILs, complement activation, kinin activity, recruitment of immune cells, and modulation of inflammation and become feasible for the tumor to evade immune elimination. The chemoresistance is closely associated with EMT due to alternative tumor-stromal interactions. The increase in invasiveness in the TME or tissue stiffness under hypoxic situations and the decrease in tumor cell adhesion can result in reduced proliferation of cancer cells, and motility and high chemoresistance with the accumulation of β-catenin [95]. The mechanism may become a potential approach to alleviating drug resistance.

Recently, some studies have proved that tumor cells can induce different levels of formation of an immunosuppressive TME by secreting varieties of degrees of exosomes with PD or PD-L1 proteins [96].

In addition, drug resistance is also closely associated with temporal TMH. The exosomes with miR-21 from CAF and cancer-associated adipocytes (CAA) can confer cancer cells chemoresistance by binding with Apoptotic protease activating factor 1 and suppress cancer apoptosis. The mobility and invasion of cancer cells can be enhanced at the same time [97].

#### 5.1.4. Epigenetic ITH Due to Metabolic Temporal ITH

Temporal ITH affecting pulmonary tumor cell metabolism performs an essential role as an obstacle to treatment for LC. In addition to the aforementioned mechanisms in metabolic ITH, some ITH in key enzymes can also contribute to pulmonary tumor metabolic ITH by inducing epigenetic aberrations. Moreover, DNA methylation patterns have been reported to always introduce certain temporal heterogeneity in advancing LC [98]. As far as other processes of epigenetic variations are concerned, spatiotemporal heterogeneity also makes a profound difference in them, including chromatin remodeling, histone acetylation, and regulation of transcription and expression of specific genes by noncoding RNAs [99]. Epigenetic ITH can emerge reversibly, along with the proliferation of lung cancer cells, the effect of which not only acts as metabolic ITH but also influences the immunogenicity by disturbing the regulation of driver gene expression (in addition, some genes taking charge of coding for antigenic objects can be covered), including that of (latent) viral proteins, TNAs, and tumor-associated antigens (TAAs) derived from abnormal expression or overexpression of wild-type proteins leading to the leakage of central tolerance. Investigation of metabolic ITH in LC essentially makes sense for the prognosis and evolution of LC after chemotherapy.

### 5.2. Spatial ITH in LC

Spatial ITH performs another indispensable role in progressing LC (Figure 7). Among different regions of the same LC, different features of TME form the pressure of cell survival and dynamic of evolution, such as regional oxygen levels and secreted factors, the representation of which is transforming growth factor beta 1 (TGFB1) [100]. Additionally, different levels of protein expression on cellular surfaces in different areas, such as PD-1 ligand CD274 (also called PD-L1), can provide the potential possibility of spatial heterogeneity [101]. Hence, the advancement of LC and response to specific treatment must be recognized in a spatial heterogeneity manner so that the prognosis of LC can have likelihood to be improved.

#### 5.2.1. Difference Related to Regions Due to Genetic Spatial ITH

In LC, with mutations on separated regional driver genes exhibiting a high spatial ITH rate, the prognosis shows worse, such as *MET* in non-squamous lung carcinomas [102]. To solve the problems brought by spatial ITH between primary survival locality and metastasis, so much research and investigations have been conducted on sequential sampling or genetic materials of pulmonary tumor cells by liquid biopsy, such as cell-free DNA, such as ctDNA and accumulating CTCs. These methods are meaningful when selecting the proper and accurate treatment, monitoring and evaluating early pulmonary tumor recurrence, resistance obtainment, and minimal residual disease [103,104,105].

#### 5.2.2. The Classification and Diversity in TME Due to Spatial ITH

Spatial ITH in TIME can present three extremely simplified models, which are, respectively called hot/cold/excluded triad. The hot pulmonary tumors are exceedingly sensitive to kinds of therapies in general, such as immune checkpoint inhibitors (ICIs) therapy, because of many CTLs. By contrast, the cold counterparts present obvious resistance to the same therapies, as T cells infiltrate TME limitedly. It is the same as excluded pulmonary tumors that the reaction of varieties of therapies is rather poor. Limited CTLs in the periphery of the TME perform an indispensable role in resistance performed by excluded counterparts because of a dense stroma or some immunosuppressive signals, which can prove that the effect of spatial ITH depends on not only quantitative consideration but also the functional cause of CTLs. In addition to CTLs, there are plenty of immune cell subtypes revealing distinguished regionally numerical and functional landscapes which have a tight connection with specific clinical parameters [106]. One of the most significant immune cells is Treg which exhibits functional ITH related to spatial ITH [107,108]. For example, when some Treg subsets perform deficient antigen presentation for being locally confined to zones, others emerge in the specific areas accelerated by CAFs.

Spatial ITH in pulmonary TME includes another important compartment called the stroma. For instance, in LC, tumor endothelial cells (TECs) attain functional ITH linked to spatial ITH. While in some regions, TECs perform original and conventional endothelial functions as primary TECs clusters, in other regions, the subsets of TECs can obtain the capability of eminent immunomodulatory activity [109].

In addition, some studies proved that the alteration of TME before metastasis has a tight connection with RNAs from tumor cellular exosomes, which can induce neutrophils into the lungs [110]. Both CAF and immune cells can behave as spatiotemporal ITH affected by exosomes from tumor cells. Meanwhile, the diversity of TME can also influence the ITH of cancer cells by exosomes from cells in TME. For example, BCC can accelerate EMT in the lung by exosomes with microRNA-210 [111] and change in pulmonary epithelial phenotype. When varieties of alveolar epithelial cells transform into AT2 alveolar epithelial cells or stem cells, ITH in TIME of pulmonary premetastatic niche directly develops towards different kinds of suitable circumstances which can benefit BCC metastasis. Moreover, exosomes with CD81 from CAFs enhance the mobility and activity of BCC by the Wnt-PCP signaling pathway [112].

## 6. ITH of LC and Novel Research Methods and Tools

Organotypic models are revolutionizing our understanding of cancer heterogeneity and its implications for personalized medicine. The models can accurately recapitulate the ITH of LC, eliminate the undesirable technical variability accompanying cancer organoid culture, and accelerate translatable insights into patient care with the establishment of reproducible platforms [113]. The integration of multiple omics datasets generated from patients using multi-omics approaches has enabled the identification of coherent and preserved molecular or clinical features across different datasets. The goal of multi-omics studies is to validate patient subgroups and biological features underlying cancer pathophysiology, understand the role of ITH in cancer genesis and development, and explore specific diagnoses and personalized therapeutic schedules. Large-scale multi-omics datasets can be used in integrative approaches to establish the connection between molecular markers and the response to targeted therapies. A more comprehensive understanding of the molecular characteristics of non-responsive or resistant tumors could lead to more precise predictions of therapy outcomes, resulting in increased therapeutic efficacy or the ability to overcome drug resistance.

ADC has been researched by CPTAC with the integration of WES, WGS, RNA-seq, miRNA, and DNA methylation profiling, and high-resolution mass spectrometry-based proteomics, phosphoproteomics, and acetylproteomics, which revealed four clusters and molecular characteristics.

A minimally invasive method called lipid biopsy allows for early diagnosis and screening, prediction of prognosis, early relapse detection in localized and locally advanced cancer, minimal residual disease (MRD) identification, and longitudinal monitoring of the disease progression and treatment response [114]. Liquid biopsy components, termed tumor circulome, including CTCs, cfRNA, ctDNA, TEPs, EVs, proteins, and metabolites, are secreted from tumor (apoptotic or necrotic) cells [115]. These tumor components present novel and minimally invasive biosources that are clinically implicated in precision medicine [116].

TMH of LC performs a critical role in pulmonary tumorigenesis, metastasis, and drug resistance. Hence, the deconvolution of TME can be essential to every aspect of LC research and treatment. With the development of novel spatially resolved high-plex molecular profiling technologies, we can obtain an in-depth understanding from TME perspectives due to their capacity to allow high-plex protein and RNA profiling while keeping valuable spatial information. Multiple available spatial proteogenomic technologies can provide the approaches to explore the dynamic interplay within the TME and elucidate mechanisms enabling the exploration of novel drugs and implementation of new biomarkers. Furthermore, TIME suggests that functional PD1/PD-L1 requires not only the expression of these proteins but also the co-localization of leukocytes. Effective and controlled immunogenic responses can only be achieved by activating immune-primed tumor cells within the TME. This activation leads to favorable antitumor effects elicited by TILs.

Hence, spatial transcriptomics and proteomics analysis are necessary to provide an unprecedented analytical scope in identifying TMH of LC and uncovering complicated mechanisms of interplay between tumor cells and immune contexts.

## 7. Therapeutic Strategies for Treating LC with Elevated ITH

As mentioned above, pulmonary tumors obtain the capability of therapy resistance and immunosuppress due to the elevation of the degree of ITH. Meanwhile, ITH offers a new potential target for treatment (Figure 8).

The natural history of LC includes rapid evolution from chemosensitivity to chemoresistance. The mechanism of the treatment resistance has a tight connection with increased ITH, including heterogeneous expression of therapeutic targets and potential resistance pathways, such as epithelial-to-mesenchymal transition, between cellular subpopulations following treatment resistance. Similarly, serial profiling of patient CTCs directly from blood confirmed increased ITH post-relapse. These findings suggest that treatment resistance in LC is characterized by coexisting subpopulations of cells with heterogeneous gene expression leading to multiple concurrent resistance mechanisms [117]. Hence, clinical efforts would be necessary to focus on rational combination therapies for the treatment of LC tumors to maximize initial responses and counteract the emergence of ITH and diverse resistance mechanisms.

Multiple therapeutic strategies have been conducted to improve the prognosis of cancer patients. For example, the precise, personalized therapy by one patient panel has been reported to provide a powerful platform for optimizing drug selection and biomarker discovery, which may speed up the emergence of precision therapeutic regimens [118]. Similarly, a case report demonstrated that the ITH introduces significant challenges in designing effective treatment strategies, and the better understanding of the ITH may provide a clinical befit for cancer patients [119]. There have been multiple attempts to study ITH and optimal strategies. Many have focused on theoretical approaches to examine drug scheduling with generic single drugs or drug combinations on a heterogeneous population containing a sensitive and resistant subpopulation [120,121]. Some of these scheduling strategies have been followed by experimental validations in vitro [122,123]. However, the generalization of the mechanisms of these multiple strategies targeting the treatment of regulating ITH was still incomplete. Therefore, we summarize the mechanisms of these strategies on how to kill tumors with elevated ITH.

Therapeutic strategies can be divided into four perspectives, including ITH level modulation, antigen spread, homogeneous antigen targeting, and multi-antigen targeting, each of which has its own benefits and limitation at the same time.

The methods of ITH level modulation mainly cover two components; one of them is boosting ITH to intolerable levels, which can elevate the degree of ITH in order to provoke immunoediting and evasion and also be effective against pulmonary tumors with high levels of genomic instability [124]. The other one is a completely inverse approach, by suppressing ITH-promoting processes or reducing ITH-promoting cell subpopulations, the pre-eminence of which can be against pulmonary tumors with epigenetic ITH by means of inhibitors of epigenetic regulators or with a highly glycolytic profile [125,126]. Both have common defects, which are the limitations of clinical therapy, including five main points. The restriction to two methods has emerged as a limitation to the neoplasms depending on ITH for resistance or progression and to those with persistently high levels of ITH. Moreover, the means can be limited by neoplasms with the potential possibility of resistance to therapy.

The next strategy in antigen spreading also covers two parts, namely, immunogenic cell death (ICD) induction with oncolytic viruses and ICD induction with radiation therapy or oncolytic peptides. As stated in previous studies, advantages of the former include the efficiency of constraining pulmonary tumors with a high degree of ITH, resulting from rather an unclear tropism, while the defects of it restrain the harness and popularity, incorporating difficulties in identifying accurate treatment schedules, problems in viral clearance, limitations of monotherapy for antitumor activity, the appearance of neutralizing antiviral antibodies, and probable immunosuppression owing to viral copying [127,128]. As far as ICD induction with radiation therapy or oncolytic peptides is concerned, two principal dominance factors encompass the improvement of recent technical accuracy by which ionizing radiation can be exactly delivered to pulmonary tumors, particularly in allusion to SCLC and high matched degree of combination with immune checkpoint inhibitors. The shortcoming can always emerge as immunosuppressive affection, limited specificity, distinguished sensitivity of TME compartments, and the likelihood of an increase in ITH of non-malignant TME compartment [129,130,131].

Methods of homogeneous antigen targeting can be divided into three parts. The first part is to exploit T cells engineered by TCR to act against clonal TNAs. It is useful for the limitation of antigen loss by targeting TNAs which are encoded by driver genes with hot-spot mutations. However, obstruction of popularization covers elevated cost, engineered T cell exhaustion, construction of fibrotic or immunosuppressive TME, scarce fully clonal TNAs/TAAs, time-assuming experimental set-up and process, the high toxicity of TAAs expressed by normal tissues, low immunogenicity of TAAs, potential TNA/TAA loss, and restriction to pulmonary tumors with capable cell death signaling and antigen presentation [132,133,134]. The second part is to harness CAR T cells and bispecific T cell engagers (BiTEs), which can improve the potency, persistence, and safety of treatment owing to high molecular flexibility. The function of limiting resistance due to antigen escape can be facilitated by the cooperation of low-dose radiation therapy. However, the drawback of this method lies in its requirement of high antigen density, other immune-related toxic effects, engineered T cell exhaustion, limited effectiveness against solid pulmonary tumors, toxicity associated with target expression by normal tissues, potential target loss, and construction of fibrotic or immunosuppressive TME [135,136,137,138]. The third part is to take advantage of tumor-specific unconventional lymphoid cells identifying nonpeptide antigens, such as lipids and small metabolites. In contrast, the limitation is more apparent, presenting as ligands not identified, the establishment of fibrotic or immunosuppressive TME, ineffectiveness against B2M-deficient tumors, demonstrated only for MR1-limited T cells, and deficiency in evidence of long-term immunological memory.

The method of another strategy depends on T cells that target multiple clonal and/or subclonal TAAs or TNAs. The main goal can be achieved by TCR- or CAR-expressing T cells, multiple monospecific T cell clones, or a single T cell clone that encodes receptors for distinguished antigens. The limitations of this strategy include exhaustion of engineered T cells, the requirement of high antigen density, construction of fibrotic or immunosuppressive TME, limited effect against solid pulmonary tumors, and toxicity related to target expression in some normal tissues [134].

The research of RCD has deepened the cognition of ITH in LC and, meanwhile, provided multiple strategies to improve LC therapy. For example, the apoptotic pulmonary tumor cells can instruct their own clearance and, at the same time, avoid the recruitment of inflammatory cells that might cause further tissue damage. In NSCLC, tumor immunosurveillance can be positively influenced by the expression of calreticulin (CRT). The surface exposure of CRT (after the execution of apoptosis) can be recognized by phagocytes through the LDL-receptor-related protein (LRP). CRT exposure can enhance the antitumor immune response resulting from an increase in the expression of vascular cell adhesion molecule (VCAM)-1 and intercellular adhesion molecule (ICAM)-1 on tumor endothelial cells [139]. Necrosis, which may occur as a result of cytotoxic therapy, can induce protective immunity. HMGB1, released during cell injury and necrosis, interacts with immune-activating receptors, such as TLR-2, -4, and -9, and the receptor for advanced glycation end products (RAGE), triggering pro-inflammatory signaling pathways [140,141]. The different levels of activated ferroptosis have a connection with ITH in cancer-relevant genes, such as RAS and TP53, in gene-encoded proteins participating in stress response pathways, and in epithelial-to-mesenchymal transition. The difficulty that we still must face is inflammation-associated immunosuppression in TME which ferroptosis damage triggers [142].

Opportunities created by adaptive responses and senescence to targeted therapies in LC can be dealt with using therapeutic resistance due to ITH [31,143].

Another potential direction of treatment is related to tumor-resident intracellular microbiota. Defense against fluid shear stress accelerates tumor metastasis induced by microbiota. Focus on this mechanism may help suppress tumor metastasis and alleviate spatial ITH in LC [144].

## 8. Conclusions

ITH in LC has been demonstrated with respect to source, mechanism, dynamic development, and therapeutic strategies, in response to negative influence, along with spatial and temporal dimensions. Previous statical opinions on pulmonary tumors can now be replaced by dynamic and developmental perspectives. That way, more theoretically feasible approaches can be put forward, and the limitations due to resistance can be overcome.

Both spatial and temporal alterations in the microenvironment of pulmonary neoplasms can enable the identification of new therapeutic targets, and the prediction of specific biomarkers. With respect to metabolic ITH, the difference in TME between malignant and non-malignant compartments can be recognized fundamentally and dynamically. Intracellular factors affecting metabolic ITH can also provide various signal pathways and genetic targets for creating new strategies for LC treatment. Moreover, correlative genetic factors not only supply primary power to launch the ITH in the perspective of metabolism or immunology but also provide an opportunity for the alteration of pulmonary tumor cells and TME due to a balance between intrinsic and extrinsic factors.

Considering the above discussion, we believe that persistent improvement of therapeutic strategy should be based not only on the exploration and excavation of plasticity and spatiotemporal heterogeneity of the pulmonary tumor but also on the deeper recognition of interaction between intrinsic and extrinsic factors affecting ITH in LC.

In this review, many new therapeutic approaches based on ITH in LC are summarized. Meanwhile, part of the potential strategies may present new mechanisms on the origination of ITH and a new relationship between mechanisms and strategies. When exploring the unknown area of the pulmonary tumor, most new findings can be explored and researched with a deeper insight of ITH in LC. At the same time, it is still a long way for us to combine the dynamic and developmental points with varieties of new mechanisms of treatment for LC.

## Figures and Tables

**Figure 1 cancers-15-02709-f001:**
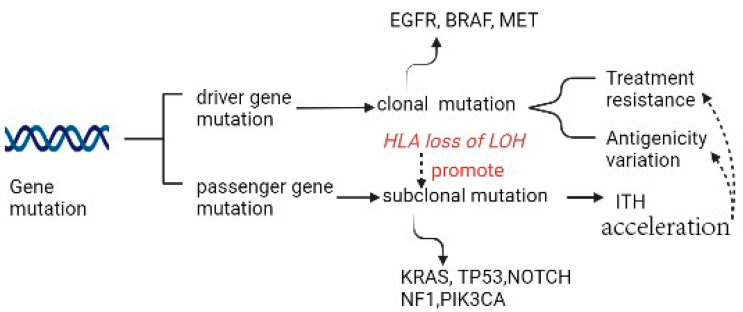
Gene mutation as the source of ITH.

**Figure 2 cancers-15-02709-f002:**
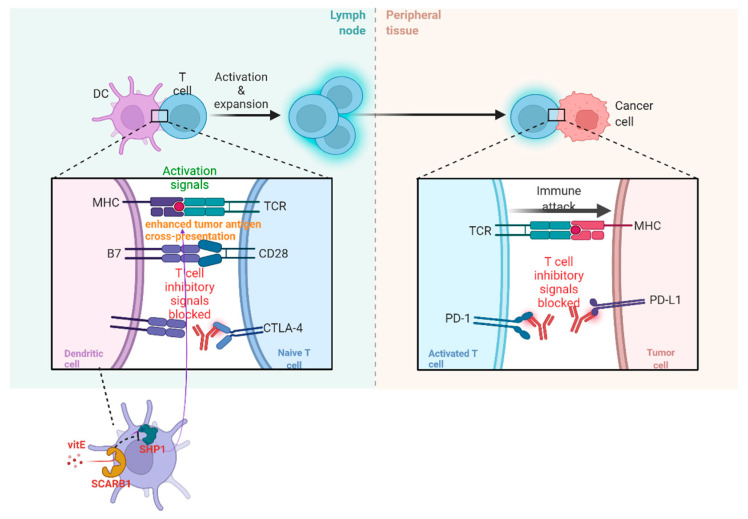
The operating principle of ICB and vitE.

**Figure 3 cancers-15-02709-f003:**
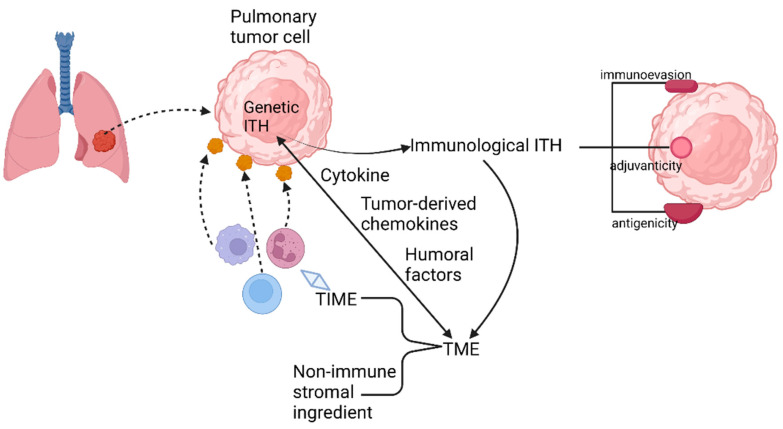
Relationship among genetic ITH, immunological ITH, and TME.

**Figure 4 cancers-15-02709-f004:**
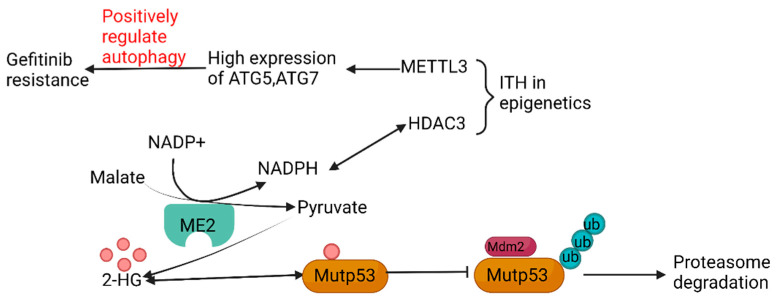
Relationship among metabolic ITH, epigenetic ITH, and genetic ITH.

**Figure 5 cancers-15-02709-f005:**
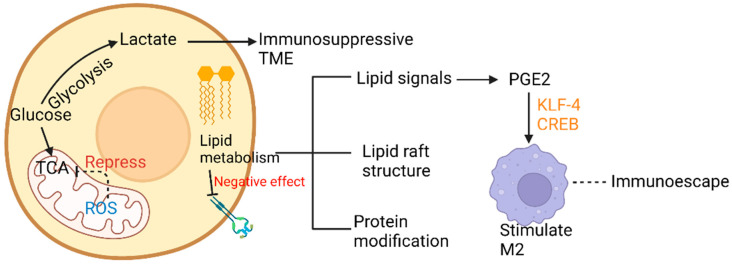
Glucose and lipid metabolic ITH affecting TME.

**Figure 6 cancers-15-02709-f006:**
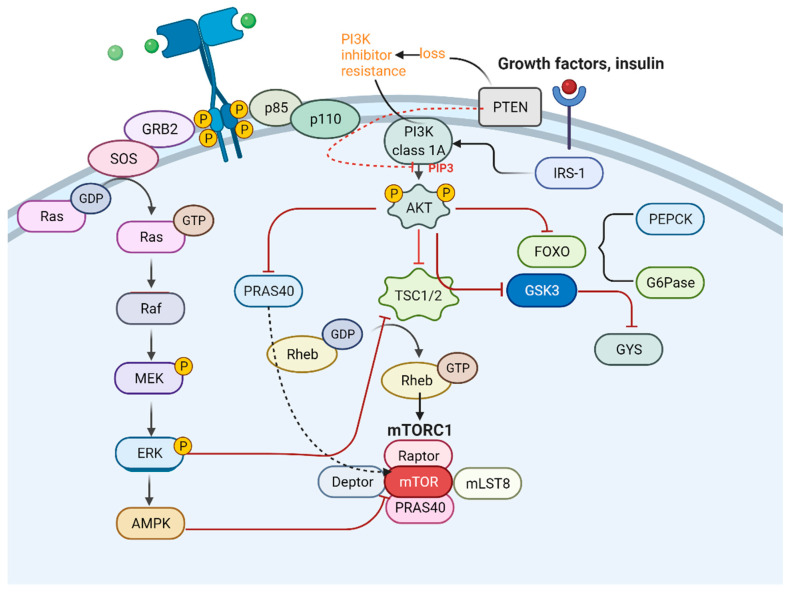
Connection between PI3K and MAPK pathway.

**Figure 7 cancers-15-02709-f007:**
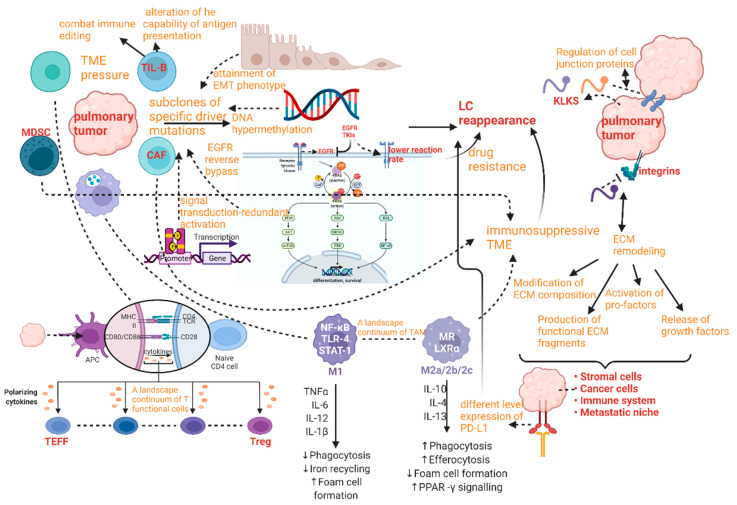
Spatial and temporal ITH in LC according to disease progression.

**Figure 8 cancers-15-02709-f008:**
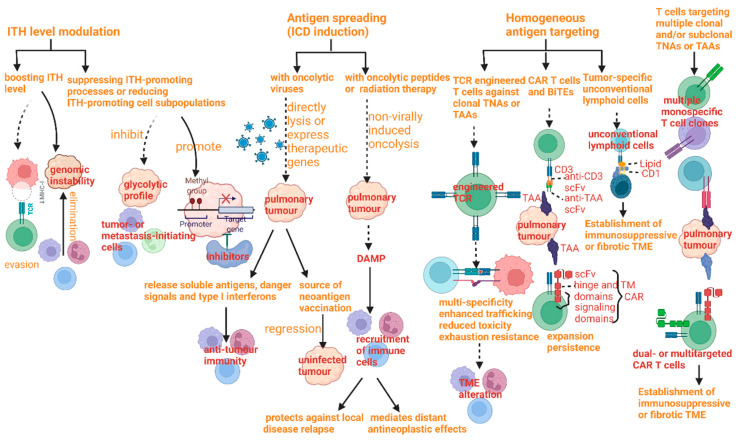
Therapeutic strategies to kill tumors with elevated intratumoral heterogeneity and their mechanisms.

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
