# Peer review of "Intratumoral Heterogeneity in Lung Cancer"

_cancers, 2023, doi:10.3390/cancers15102709_

Round 1
Reviewer 1 Report
This is a relevant and well elaborated review in the context of intratumoral and microenvironmental heterogeneity, respectively ITH and TME, with future implications for personalized lung cancer therapy. The review addresses spatial and temporal alterations in the TME of lung tumors capable of recognizing new therapeutic targets, as well as the prediction of specific biomarkers. Interestingly, the authors address that intracellular factors affecting metabolic ITH can also provide various signal pathways and genetic targets for creating new strategies of lung cancer treatment. They also address correlated genetic factors with primary capacity to initiate the ITH regarding metabolism and immune landscape, but also provide a chance for modification of lung tumor cells and TME due to a balance between intrinsic and extrinsic factors.
The authors also explore the source of lung cancer ITH, such as gene mutation which represents the basis of spatiotemporal ITH in TME and metabolism in lung cancer.
The text is very clear with adequate figures to illustrate the mechanistic in tumor cells and their microenvironment.
Since the authors are emphasizing the ITH in TME, my only concern with the review is the absence of comments about the extra matricellular (ECM) components that provides the cells with histoarchitectural support and anchoring. The ECM is composed of a complex network of highly cross-linked components, including fibrous proteins, glycoproteins, proteoglycans, and polysaccharides. The biomechanical and biochemical properties of the ECM regulate cell survival, proliferation, differentiation, and motility through the action of proteoglycans proteins. The molecular changes that occur in the ECM have been potentially associated with invasive carcinoma. Furthermore, the modifications undergone by the ECM can modulate important signaling pathways in tissue morphogenesis, and immune landscape. Theefore, I suggest to the authors to include a brief comment about their experience or the literature review about these important components of the ECM.
Thank you for give me the opportunity to review this manuscript and learn more about Intratumor Heterogeneity.
Author Response
Dear reviewer 1,
Thank you so much for your time and effort that you dedicated to providing feedback on our manuscript. The insightful comments are valuable for improving the paper. We have answered each of your points below.
We hope the revised version can be suitable publication. If the revised version is still unsatisfactory, please send back to us for further modification. Look forward to hearing from you in due course.
I suggest to the authors to include a brief comment about their experience or the literature review about these important components of the ECM.
The literature review about the important components of the ECM has been added in 5.1.3 section and highlighted in green in the revised manuscript as follows:
It has been proven that modification of tumor-stromal interactions plays an indispensable role in promoting temporal TMH where extracellular matrix (ECM) remodeled (termed as desmoplastic stroma) with the cleavage of adhesion proteins such as integrins and cadherin which can regulate migration and attachment of cancer cells and with the processing of intercellular junction proteins (desmoglein (Dsg)/desmocollin (Dsc)) which can control tumor cohesion and with the alteration of intercellular signaling. ECM proteins can be modulated by the regulation of plasmin, MMPs, uPA and meprins operated by kallikrein-related peptidases (KLKs) and promote the modification of ECM composition, production of angiostatin, activation of pro-factors (HGF, EGF and TGF-β) and release of growth factors. TMH occurs with the temporal variation of CAFs, immune cells and endothelial cells in the primary tumor site due to the ECM remodeling with the changes to ECM cross-linking and tissue stiffness mediated by tumor secreting KLKs [89, 90]. ECM can also affect ITH from perspective of regulating the tumor migration ability. When cancer cells migrate through ECM, the latter acts as a mechanical barrier to discern differences in physical properties of cells such as their size, shape, and rigidity. The collective tumor cells invasion involves ensembles of cells moving as ‘leaders’ and ‘followers’ and has been proposed to require greater cell–cell adhesion and matrix degradation as requisites for migration. However, even mesenchymal cells, with decreased expression of the epithelial markers such as E-cadherin are known to migrate collectively [91]. Cancer cells show different behaviors in distinct ECM milieu. For example, the response of cancer cells to chemotherapy has been reported to be regulated by substrata of different ECM molecules or their combinations (including, but not limited to collagen I, IV and VI, laminin, and fibronectin). In addition, apoptosis of cancer cells can be modulated by ECM in a FAK- and β1 integrin-pMLC-YAP signaling-dependent manner [92]. Tumor migration and chemoresistance are sensitive to the nanospacing of the ECM ligands that engage their cognate receptors on the cell surface, suggesting that such plasticity is linked to dynamical changes in signaling landscapes [93].
Reviewer 2 Report
Report
Review article “Intratumoral heterogeneity in lung cancer” submitted by Xiaodi Lv et al describes multiple factors contributing to spatio-temporal heterogeneity in the microenvironment of pulmonary neoplasm that contribute to identification of novel therapeutic targets as well as predictive or prognostic biomarkers. Authors have summarized novel therapeutic approaches based on intratumoral heterogeneity in lung cancer. Review is elaborate and extensive. Nevertheless, authors might want to update based on recent paper published by Wu HJ et al 2022. Minor mistakes like typo- line 350 may be corrected.
https://www.sciencedirect.com/science/article/pii/S2666979X22001070.
Author Response
Dear reviewer 2,
Thank you so much for your time and effort that you dedicated to providing feedback on our manuscript. The insightful comments are valuable for improving the paper. We have answered each of your points below.
We hope the revised version can be suitable publication. If the revised version is still unsatisfactory, please send back to us for further modification. Look forward to hearing from you in due course.
1. Authors might want to update based on recent paper published by Wu HJ et al 2022. Minor mistakes like typo- line 350 may be corrected.
The mistakes have been corrected. And the updated content has been added in section 6 according to the recent paper published by Wu HJ et al 2022 highlighted in orange in the revised manuscript as follows:
A minimally invasive method called lipid biopsy allows for early diagnosis and screening, prediction of prognosis, early relapse detection in localized and locally advance cancer, minimal residual disease (MRD) identification, and longitudinal monitoring of the disease progression and treatment response [115]. Liquid biopsy components, termed tumor circulome, including CTCs, cfRNA, ctDNA, TEPs, EVs, proteins, and metabolites, are secreted from tumor (apoptotic or necrotic) cells [116]. These tumor components present novel and minimally invasive biosources that are clinically implicated in precision medicine [117].
Reviewer 3 Report
In the current review authors highlight the importance of understanding ITH in the context of lung cancer and how it impacts various aspects of the disease, including diagnosis, treatment, and therapeutic resistance. Additionally, it emphasizes the need for a multidisciplinary approach to accurately address the complexity of ITH in lung cancer and improve patient outcomes.Overall it is a very good review to read and I have enjoyed reading it. I have few suggestions that can significantly improve the review for readers.- I would recommend for authors to include a section about how the treatment affects ITH, as a result the tumors have increased plasticity and how it affects overall response.
- I would also recommend authors to add the clinical trials that are underway which will potentially alter the ITH for better therapeutic response.
The English is good for readers to understand.
Author Response
Dear reviewer 3,
Thank you so much for your time and effort that you dedicated to providing feedback on our manuscript. The insightful comments are valuable for improving the paper. We have answered each of your points below.
We hope the revised version can be suitable publication. If the revised version is still unsatisfactory, please send back to us for further modification. Look forward to hearing from you in due course.
1.
I would recommend for authors to include a section about how the treatment affects ITH, as a result the tumors have increased plasticity and how it affects overall response. I would also recommend authors to add the clinical trials that are underway which will potentially alter the ITH for better therapeutic response.
The generalization of the mechanism about how the treatment affects ITH and the clinical trials that may potentially alter the ITH have been added in the “section 7” and highlighted in blue in the revised manuscript as follows:
The natural history of LC includes rapid evolution from chemosensitivity to chemoresistance. The mechanism of the treatment resistance has the tight connection with increased ITH including heterogeneous expression of therapeutic targets and potential resistance pathways, such as epithelial-to-mesenchymal transition, between cellular subpopulations following treatment resistance. Similarly, serial profiling of patient CTCs directly from blood confirmed increased ITH post-relapse. These findings suggest that treatment resistance in LC is characterized by coexisting subpopulations of cells with heterogeneous gene expression leading to multiple, concurrent resistance mechanisms [115]. Hence, the clinical efforts would be necessary to focus on rational combination therapies for treatment- LC tumors to maximize initial responses and counteract the emergence of ITH and diverse resistance mechanisms.
Multiple therapeutic strategies have been conducted to improve prognosis of cancer patients. For example, the precise personalized therapy by one patient panel has been reported to provide a powerful platform for optimizing drug selection and biomarker discovery, which may speed up the emergence of precision therapeutic regimens [116]. Similarly, a case report demonstrated that the ITH introduces significant challenges in designing effective treatment strategies and the better understanding of the ITH may provide a clinical befit for the cancer patients [117]. There have been multiple attempts to study ITH and optimal strategies. Many have focused on theoretical approaches to examine drug scheduling with generic single drugs or drug combinations on a heterogeneous population containing a sensitive and a resistant subpopulation [118, 119]. Some of these scheduling strategies have been followed by experimental validations in vitro [120, 121]. However, the generalization of the mechanisms of these multiple strategies targeting on the treatment of regulating ITH was still incomplete. Therefore, we summarize the mechanisms of these strategies on how to kill tumors with elevated ITH.